# Vaginal Microbiota of the Sexually Transmitted Infections Caused by *Chlamydia trachomatis* and *Trichomonas vaginalis* in Women with Vaginitis in Taiwan

**DOI:** 10.3390/microorganisms9091864

**Published:** 2021-09-02

**Authors:** Shu-Fang Chiu, Po-Jung Huang, Wei-Hung Cheng, Ching-Yun Huang, Lichieh Julie Chu, Chi-Ching Lee, Hsin-Chung Lin, Lih-Chyang Chen, Wei-Ning Lin, Chang-Huei Tsao, Petrus Tang, Yuan-Ming Yeh, Kuo-Yang Huang

**Affiliations:** 1Graduate Institute of Pathology and Parasitology, National Defense Medical Center, Taipei 114, Taiwan; A3183@tpech.gov.tw (S.-F.C.); kelly19960422@gmail.com (C.-Y.H.); 1100sun@pchome.com.tw (H.-C.L.); 2Department of Inspection, Taipei City Hospital, Renai Branch, Taipei 114, Taiwan; 3Department of Biomedical Sciences, Chang Gung University, Taoyuan 333, Taiwan; pjhuang@mail.cgu.edu.tw; 4Genomic Medicine Core Laboratory, Chang Gung Memorial Hospital, Linkou, Taoyuan 333, Taiwan; chichinglee@cgu.edu.tw; 5Department of Medical Laboratory Science, College of Medicine, I-Shou University, Kaohsiung 824, Taiwan; whcheng@isu.edu.tw; 6Molecular Medicine Research Center, Chang Gung University, Taoyuan 333, Taiwan; julie.chu@mail.cgu.edu.tw; 7Liver Research Center, Chang Gung Memorial Hospital, Linkou, Taoyuan 333, Taiwan; 8Graduate Institute of Biomedical Sciences, College of Medicine, Chang Gung University, Taoyuan 333, Taiwan; 9Department of Computer Science and Information Engineering, Chang Gung University, Taoyuan 333, Taiwan; 10Division of Clinical Pathology, Department of Pathology, Tri-Service General Hospital, National Defense Medical Center, Taipei 114, Taiwan; 11Department of Medicine, Mackay Medical College, New Taipei 252, Taiwan; lihchyang@mmc.edu.tw; 12Graduate Institute of Biomedical and Pharmaceutical Science, Fu Jen Catholic University, New Taipei 242, Taiwan; 081551@mail.fju.edu.tw; 13Department of Medical Research, National Defense Medical Center, Tri-Service General Hospital, Taipei 114, Taiwan; changhuei@gmail.com; 14Department of Microbiology and Immunology, National Defense Medical Center, Taipei 114, Taiwan; 15Molecular Regulation and Bioinformatics Laboratory, Department of Parasitology, College of Medicine, Chang Gung University, Taoyuan 333, Taiwan; petang@mail.cgu.edu.tw

**Keywords:** *Chlamydia trachomatis*, *Trichomonas vaginalis*, *Neisseria gonorrhoeae*, sexually transmitted infections, vaginal microbiome

## Abstract

The three most common sexually transmitted infections (STIs) are *Chlamydia trachomatis* (CT), *Neisseria gonorrhoeae* (GC) and *Trichomonas vaginalis* (TV). The prevalence of these STIs in Taiwan remains largely unknown and the risk of STI acquisition affected by the vaginal microbiota is also elusive. In this study, a total of 327 vaginal swabs collected from women with vaginitis were analyzed to determine the presence of STIs and the associated microorganisms by using the BD Max CT/GC/TV molecular assay, microbial cultures, and 16S rRNA sequencing. The prevalence of CT, TV, and GC was 10.8%, 2.2% and 0.6%, respectively. A culture-dependent method identified that *Escherichia coli* and *Streptococcus agalactiae* (GBS) were more likely to be associated with CT and TV infections. In CT-positive patients, the vaginal microbiota was dominated by *L. iners*, and the relative abundance of *Gardnerella vaginalis* (12.46%) was also higher than that in TV-positive patients and the non-STIs group. However, *Lactobacillus* spp. was significantly lower in TV-positive patients, while GBS (10.11%), *Prevotella bivia* (6.19%), *Sneathia sanguinegens* (12.75%), and *Gemella asaccharolytica* (5.31%) were significantly enriched. Using an in vitro co-culture assay, we demonstrated that the growth of *L. iners* was suppressed in the initial interaction with TV, but it may adapt and survive after longer exposure to TV. Additionally, it is noteworthy that TV was able to promote GBS growth. Our study highlights the vaginal microbiota composition associated with the common STIs and the crosstalk between TV and the associated bacteria, paving the way for future development of health interventions targeting the specific vaginal bacterial taxa to reduce the risk of common STIs.

## 1. Introduction

Vaginitis is a common disease for women with vaginal symptoms, including abnormal discharge, odor, irritation, dysuria, itching, or burning [1]. The most common causes of vaginitis are bacterial vaginosis (BV), vulvovaginal candidiasis (VVC), and trichomoniasis, accounting for approximately 40–50%, 20–25%, and 15–20% of vaginitis cases, respectively [2]. BV is characterized by decreased abundance of *Lactobacillus* spp. and increased abundance of anaerobic bacteria, such as *Atopobium vaginae*, *Gardnerella vaginalis*, *Mobiluncus mulieris*, *Prevotella* spp., BV-associated bacterim-2 (BVAB-2), and *Megasphaera* spp. [3,4]. VCC is generally caused by an infection with the fungus *Candida albicans* [5]. Trichomoniasis is the most common non-viral sexually transmitted infection (STI) caused by the protozoan parasite *Trichomonas vaginalis*. Co-infection of two pathogens occurs frequently in women with vaginitis. For instance, women with BV are potentially co-infected with *Candida* spp. and *T. vaginalis*, with co-infection rates of 20–30% and 60–80%, respectively [6]. More research is necessary to better characterize the cause of inflammatory vaginitis, which is helpful to specifically develop therapies for complete eradication.

The three most common curable STIs are *Chlamydia trachomatis* (CT), *Neisseria gonorrhoeae* (GC) and *T. vaginalis* (TV). It is estimated that the numbers of new cases worldwide for CT, GC, and TV infections are 78, 131, and 142 million every year, respectively [7]. The outcomes of these infections include pelvic inflammatory disease (PID), miscarriage, infertility, low birth weight, and increased risk of HIV transmission [8]. Increased resistance of *T. vaginalis* to metronidazole has emerged as a highly problematic public health issue [9]. Similarly, elevated antimicrobial resistance in GC-infected patients is another urgent threat in the 21st century [10]. Although true antimicrobial resistance of CT is rare [11], concern about treatment failure has arisen due to high repeat CT infections observed in several countries [12,13]. Hence, it is essential to discover alternative approaches to prevent and treat these common STIs.

*Lactobacillus* spp. primarily dominates a normal vaginal microbiome [14], producing lactic acid to maintain a normal vaginal pH and play a role in preventing STI acquisition. Human vaginal microbiomes can be subdivided into five distinct community state types (CSTs) [15], four of which (CST I–III and CST-V) are often dominated by a specific *Lactobacillus* species. CST-IV is characterized by reduced *Lactobacillus* spp. and, instead, consists of a higher proportion of facultative and strict anaerobes associated with BV [16]. It has been shown that CT-infected women are more likely to have cervicovaginal microbiota (CVM) dominated by *L. iners* or by diverse anaerobic bacteria, than by *L. crispatus* [17]. Additionally, specific vaginal bacteria, such as *Prevotella*, *Sneathia*, *Mycoplasma*, and *Parvimonas*, are associated with TV acquisition [18,19]. A previous study suggests that members of the CVM can modify *N. gonorrhoeae*, potentially facilitating the infectious process to men [20]. Based on the microbiome analyses associated with the common STIs, it is feasible to develop novel vaginal health interventions targeting specific bacterial species, and thus, reduce the risk of STI acquisition.

To date, the true prevalence of CT, TV, and GC infection is unknown in Taiwan, which may lead to mixed vaginitis and many co-infections. It is still elusive whether specific vaginal bacterial species or overall microbiota diversity contributes to an increased risk of these common STIs acquisitions. In this study, we used the BD max CT/GC/TV (MAX) panel assay for the diagnosis of CT, TV, and GC from samples initially collected for diagnosis of vaginitis. A cultivation-based identification of the STI-associated bacteria by MALDI-TOF analysis was utilized. We also conducted culture-independent 16S rRNA sequencing to investigate the relationships between the common STIs and the composition of the vaginal microbiota. Specific bacteria taxa enriched from the microbiome analysis were further co-cultured with TV to verify the bacteria–parasite interactions.

## 2. Materials and Methods

### 2.1. Sample Collection and Ethical Approval

A total of 327 vaginal swabs were collected from women (age 18–80 years old) diagnosed with vaginitis who sought medical care in an emergency department, Obstetrics and Gynecology (OB/GYN), and Sexually Transmitted Diseases (STD) Prevention Center in Northern Taiwan. Participant selection for this study took place between September 2018 and March 2019. Samples were identified only with the study identification number that had been assigned during the BD Max assay. This study was approved by the Committees for Ethics of Taipei City Hospital, Renai Branch (TCHIRB-1083008-E), with permission to use residual samples.

### 2.2. Culture-Dependent Identification of Microorganisms by MALDI-TOF Analysis

Vaginal swabs for culture were placed in tubes containing Amies to maintain the swabs’ moisture and prevent autolysis of the microorganisms. The vaginal specimens were plated onto 5% sheep blood agar, chocolate agar, and eosin methylene blue medium. The culture plates and medium were incubated at 37 °C aerobically for 24 h. Identification of the bacterial species was carried out by a phenotypic method, Gram stains, and MALDI-TOF analysis.

### 2.3. BD Max CT/GC/TV Assay and DNA Extraction

For diagnosis of the three most common STIs in women, the BD Max CT/GC/TV platform (BD, Franklin Lakes, NY, USA) provides high sensitivity and specificity using vaginal swabs [21]. Briefly, the BD Max assay is a TaqMan-based PCR assay that uses target-specific primers and probes to perform amplification and detection of amplified products, providing a diagnosis of chlamydia, gonorrhea, and trichomoniasis simultaneously. DNA extraction, reagent rehydration, amplification, and target nucleic acid sequence detection were handled automatically by the BD Max instrument (BD, USA).

### 2.4. Analysis of Microbiota Composition

#### 2.4.1. PCR Amplification and Sequencing

After DNA extraction from the BD Max system, the amplicon library was constructed by PCR targeting the 16S rRNA V3-V4 region as previously described [22]. Illumina adaptor overhang nucleotide sequences were added to the gene-specific sequences (16S rRNA gene amplicon PCR forward primer sequence 5′-TCGTCGGCAGCGTCAGATGTGTATAAGAGACAGCCTACGGGNGGCWGCAG-3′ and 16S rRNA gene amplicon PCR reverse primer sequence 5’-GTCTCGTGGGCTCGGAGATGTGTATAAGAGACAGGACTACHVGGGTATCTAATCC-3’) [23]. The first PCR mixture contained 1 µM of forward and reverse primers, 1× KAPA HiFi HotStart ReadyMix (Roche, Brighton, MA, USA), and 10 ng genomic DNA. The first PCR consisted of 3 min of denaturation at 95 °C followed by 25 cycles of denaturation at 95 °C/30 s, annealing at 55 °C/30 s, and extension at 72 °C/30 s, with a final 72 °C extension for 30 s. The PCR products were purified using Agencourt AMPure XP beads (Beckman Coulter, Brea, CA, USA) and subjected to index PCR. Each index PCR mixture contained 5 µL of both Nextera™ XT index set primers (Illumina, San Diego, CA, USA), 1× KAPA HiFi Hotstart Ready Mix, and the purified products of the first PCR. The index PCR consisted of 3 min of denaturation at 95 °C followed by 8 cycles of denaturation at 95 °C/30 s, annealing at 55 °C/30 s, and extension at 72 °C/30 s, with a final extension at 72 °C for 5 min. The final amplicon libraries were approximately 630 bp in length and were validated using the Agilent 2100 Bioanalyzer with the Agilent HS DNA Kit. The sequencing of the multiplexed pooled libraries was performed on the MiSeq System with MiSeq Reagent Kit v3 (600 cycles) (Illumina, CA, USA).

#### 2.4.2. Bioinformatics Analysis

The sequencing reads were initially de-multiplexed using MiSeq Reporter v2.6 according to the sample barcodes. The resulting pairs of reads were processed by using USEARCH (v11) (https://drive5.com/ (accessed on 27 July 2021)). Briefly, we used the *fastq_mergepairs* command to merge the paired reads and the *fastq_filter* command to filter low quality reads by setting a maximum number of expected errors (E_max) = 1. The average merged reads of samples were 72,618 ± 15,482 tags. The Cutadapt v2.3 package [24] was used to remove forward and reverse sequencing primers from the merged reads of each dataset. Only sequence tags with length ≥ 400 bp were retained for subsequent analysis. The de-replication reads from effective reads were denoised to identify all correct biological sequences clustered into a zero-radius operational taxonomic unit (zOTU) using the UNOISE3 [25]. Furthermore, the RDP training set (v16) served as a reference species database and the final taxonomic assignments were performed using the SINTAX algorithm [26]. The α-diversity (e.g., Chao1 index and Shannon index) and β-diversity (e.g., Bray–Curtis dissimilarity) normalized by DESeq2 [27] were calculated by Qiime (version 1.9.1) [28]. Canonical correspondence analysis (CCA) and constrained principal coordinate analysis (CPCoA) were used to visualize the data in R (version 3.6.3). Finally, we used linear discriminant analysis (LDA) of effect size (LEfSe) [29] to determine the taxonomy that explains the significant differences in biomarkers between STI-positive and non-STI samples. The Kruskal–Wallis (KW) sum-rank test and pairwise Wilcoxon rank-sum test were used to determine the LDA score, with which greater than 3 is considered significant.

### 2.5. In Vitro Interactions of Specific Bacteria with TV

GBS and *L. iners* were cultured in Columbia agar II, containing 8% whole horse blood at 37 °C and 5% CO_2_. TV (ATCC 30236) was maintained in YIS medium [30], pH 5.8, containing 10% heat-inactivated horse serum and 1% glucose at 37 °C. The co-incubation of TV with the associated bacteria was performed as previously described [31]. Briefly, TV (5 × 10^5^ cells/mL) and bacteria were co-incubated in 1 mL of keratinocyte serum-free media (K-SFM) (Invitrogen, Carlsbad, CA, USA) at a ratio of 1:10 at 37 °C for 1, 2, 4, and 6 h. Incubations of TV and bacteria alone were performed in parallel as controls. After co-incubation, the microbial cultures were plated out on agar plates for 24 h to monitor the growth of bacteria. Spread plates were photographed, and CFU counts were obtained. To observe the effect of co-incubation on TV growth, the microbial cultures were centrifuged, washed, and resuspended in YIS medium. The growth of the parasites was monitored by using trypan blue exclusion hemocytometer counts.

### 2.6. Statistical Analyses

Quantitative data were expressed as mean ± SD of three independent experiments. The unpaired Student *t*-test (two-tailed) was used to evaluate the significant differences between the two groups. Values of *p* < 0.05 were considered statistically significant (*) and *p* < 0.01 (**) or *p* < 0.001 (***) to be very statistically significant.

## 3. Results

### 3.1. The Prevalence of STIs and Their Differential Diagnosis in Women with Vaginitis

Three hundred and twenty-seven vaginal swabs specimens were originally collected from women with vaginitis who visited hospitals in Northern Taiwan, of which 323 were eligible for the BD Max assay. There were 48 STI-positive cases among the vaginitis patients. The demographic characteristics, clinic type, and differential diagnosis of each STI are summarized in Table 1. The prevalence of CT was 10.8% (35/323), TV was 2.2% (7/323) and GC was 0.6% (2/323). The mean age for the participants with CT, TV, and GC infection was 31.6, 37.1, and 25.2 years, respectively. Notably, there was a higher proportion for older women in the TV-positive group, with 57% of TV-infected patients above 40 years, whereas younger women were predominant in the CT- and GC-positive groups. In the CT-positive cases with various symptoms, there was a higher proportion of PID (*n* = 7, 20%). In the TV-positive cases, subacute vaginitis (*n* = 2, 28.5%) and cervicitis (*n* = 2, 28.5%) were the predominant symptoms. There were four participants with mixed STIs, including two with CT/TV, one with TV/GC, and one with CT/TV/GC (Appendix A). In summary, we firstly used a molecular-based test to evaluate the prevalence of CT, TV, and GC infection in women with vaginitis in Taiwan, and found that CT is the most common infection among these STIs. 

### 3.2. Identification of Bacterial Co-Infections in STI-Positive Patients by a Culture-Dependent Method Combined with MALDI-TOF Analysis

It has been demonstrated that traditional culture-dependent methods and modern molecular techniques should be performed simultaneously as their combined results would reflect the bacterial community composition more precisely [32]. Hence, we initially used a culture-dependent method combined with MALDI-TOF analysis to identify the specific microorganisms co-infected in the vagina of STI-positive patients (Figure 1). Since the sample size of GC-positive and mixed STIs was limited, we herein focused on the bacteria co-existing in the CT- and TV-positive samples. It is noteworthy that GBS is the most prevalent bacteria co-infected with these STIs, either single (CT or TV) or mixed infections (CT/TV, TV/GC, and CT/TV/GC). In the CT-positive samples, GBS (*n* = 8), *Escherichia coli* (*n* = 8), and *Staphylococcus* spp. (*n* = 7) were the main co-existing bacteria. The overgrowth of *Candida* spp. or *G. vaginalis* in the genital ecosystem may lead to microbial dysbiosis that increases the risk of acquiring CT [33]. Similarly, *C. albicans* (*n* = 2) and *G. vaginalis* (*n* = 2) were also identified in CT-infected patients. Additionally, GBS (*n* = 2), *E**. coli* (*n* = 2), *Klebsiella pneumoniae* (*n* = 1), *Staphylococcus* spp. (*n* = 1), *Enterococcus faecalis* (*n* = 1), and *G. vaginalis* (*n* = 1) were identified as being associated with TV infections. Together, microbial culture results revealed that several bacterial species co-exist in the CT- and TV-positive samples, especially GBS and *E. coli*, whereas *Lactobacillus* spp. and *C. albicans* are only identified in the CT- but not TV-positive samples.

### 3.3. Vaginal Microbiota Profiling in STI-Positive Patients Using 16S rRNA Sequencing

To characterize the vaginal microbiome community in STI-positive patients, vaginal swabs were collected from CT-positive patients (*n* = 22), TV-positive patients (*n* = 7), GC-positive patients (*n* = 2), mixed infections (TV/CT = 2; TV/CT/GC = 1), and non-STI controls (NC, *n* = 36). The bacterial DNA was obtained from the samples via the BD Max system, and the bacteria 16S rRNA gene amplicon library was constructed targeting on the V3–V4 regions. The vaginal microbiota profiling in the STI-positive and NC samples were investigated using 16S rRNA sequencing. Microbial community variation among STI-positive and NC groups was determined by Chao1 (species richness) and Shannon (species diversity) indices and principal coordinate analysis (PCoA). We found that the presence of TV was more likely to have higher richness and diversity of the vaginal microbiota compared with the NC- and CT-positive groups (Figure 2A,B). PCoA revealed that the beta diversity of each CT- and TV-positive sample as well as the NC cases tended to cluster separately while the GC-positive case was more close to CT-positive patients. Additionally, the mixed infections (TV/CT and TV/CT/GC) showed similar bacterial community composition to the CT- and TV-positive groups (Figure 2B). These results suggested that the vaginal bacterial community was different between the STI-positive and NC groups.

### 3.4. Vaginal Microbiota Revealed Specific Bacterial Taxa Associated with STI-Positive Patients

To identify the differentially enriched bacterial taxa within the STI-positive and NC groups, linear discriminant analysis (LDA) was performed. The cladogram showed differences in 12 taxa among TV, GC, and mixed infections (TV/CT and TV/CT/GC) groups (Figure 3A). Bacteria from the *Chlamydiae* phylum and the *Betaproteobacteria* class were indeed identified in the TV/CT/GC and GC groups, respectively. The histogram displayed LDA scores of microbial taxa with significant differences within groups, showing that 19 bacterial taxa were enriched in the STI-positive groups (Figure 3B). The phylum *Tenericutes* was significantly abundant in the TV-positive group. The genus *Prevotella* was specifically correlated with the TV/CT group, while *Mycoplasma* and *Gemella* were associated with the TV-positive group. Taxonomic profiles of the bacterial community at the genus level revealed that *Lactobacillus* dominated in CT-positive patients as well as in the NC group, whereas it was specifically reduced in TV-positive patients (Figure 3C and Appendix A). Specifically, *Gardenella* spp. was found to be more abundant in the CT-positive group compared with the TV-positive and NC groups. This result was consistent with the finding identified by the aforementioned culture-dependent method. Additionally, *Streptococcus* spp., *Prevotella* spp., and *Sneathia* spp. were strongly associated with TV-infected patients compared with the NC group. *Bifidobacterium breve* has been shown to exert a strong antimicrobial activity against urogenital pathogens and significantly reduces CT invasion to HeLa cells [34]. We found that *Bifidobacterium* spp. was less abundant in CT- and TV-positive patients compared with the NC group. We further highlighted the enriched bacteria at the species level. *L. iners* (30.94%) and *G. vaginalis* (12.46%) were specifically abundant in the CT-positive group, whereas GBS (10.11%), *P. bivia* (6.19%), *S. sanguinegens* (12.75%), and *G. asaccharolytica* (5.31%) were enriched in the TV-positive group (Figure 3D and Appendix A). Additionally, the probiotic *B. breve* (0.88%) was distinctly enriched in the non-STI group. Collectively, our microbiome analysis indicated that specific vaginal bacteria showed significantly different abundance in CT- and TV-positive patients and generated a useful bacterial panel at the species level in these STIs compared to non-STI individuals.

### 3.5. In Vitro TV-Bacteria Co-Culture Revealed Different Influence on the Growth of L. iners and GBS

As *L. iners* was significantly reduced in the vaginal microbiota of TV-positive patients, we performed a co-culture assay to evaluate the effect of TV on bacterial growth. TV was co-incubated with *L. iners* at a ratio of 1:10 for different time intervals (1, 2, 4, 6 h) and the growth of *L. iners* was monitored by CFU after 24 h. Short-term interaction of bacteria and TV within 2 h showed that the growth of *L. iners* was markedly inhibited compared with *L. iners* cultivation alone (*p* < 0.001), with a reduction of 54.2% in CFUs (Figure 4A). After long-term co-incubation for 6 h, the inhibitory effect of TV on the growth of *L. iners* was not significant. This suggests that the survival of *L. iners* was suppressed at the initial stage of interaction with TV, but the bacteria may adapt and survive after prolonged exposure to TV. Additionally, both the culture-dependent and culture-independent methods identified GBS to be enriched in the TV-positive cases. We thus evaluated the impact of TV–GBS interaction on the growth of GBS by using the same approach. It is noteworthy that the interaction between TV and GBS, regardless of the co-incubation time, enhanced the growth of GBS after 24 h compared with GBS cultivation alone (*p* < 0.001) (Figure 4B). Hence, it is postulated that TV infection potentially enhances the colonization of GBS in the vaginal ecosystem.

### 3.6. Impact of the Crosstalk between TV and the Associated Bacteria on the Growth of Parasites

On the other hand, the influence of the interactions between the specific bacteria and TV on the growth of parasites was also verified. Co-incubation of TV and *L. iners* and GBS for 6 h in the co-culture medium was performed, followed by inoculation of the microbial cultures into the TV-specific culture medium to monitor the survival of TV after 24 h. Interestingly, the growth of TV was enhanced upon interaction with *L. iners* (*p* < 0.001), with about 10-fold higher cell density of the parasites at the initial stage (0 h) than TV cultivation alone (Figure 5A). After 24 h of cultivation, the growth of TV interacting with *L. iners* further enhanced compared with TV cultivation alone (*p* < 0.01) (Figure 5A). Additionally, the co-incubation of TV and GBS for 6 h significantly reduced the survival of TV after 24 h cultivation (*p* < 0.001) compared with TV without exposure to GBS (Figure 5B). This observation, in combination with the previous data, indicates that the crosstalk between GBS and TV promotes the growth of bacteria but inhibits the survival of parasites.

## 4. Discussion

According to a Taiwan Centers for Disease Control report, approximately 4500 gonorrhea cases were confirmed in 2019. Since chlamydia and trichomoniasis are not notifiable diseases in Taiwan, the true prevalence of these STIs is still unclear. Based on a previous epidemiological study, the overall prevalence of CT infection among patients visiting sexually transmitted disease and genitourinary clinics was around 18.4% [35]. In this study, we have conducted the first molecular-based testing to assess these common STIs in women with vaginitis in Taiwan, indicating that the prevalence for CT, TV, and GC was 10.8%, 2.2% and 0.6%, respectively. We found a significantly higher proportion of CT-positive cases than TV, which is different from the trends of STIs observed in other studies using the same diagnostic method. A previous report evaluated the prevalence of these STIs in women in the United States, with positive CT, TV, and GC rates of 7.1%, 13.5%, and 2.3% [21]. Another study assessed these STIs in women with vaginitis, showing positive CT, TV, and GC rates of 6.1%, 12.3%, and 1.8% [36]. The higher prevalence of CT-positive cases than TV in Taiwanese women with vaginitis may be due to different races and ethnicity. Hence, understanding the vaginal microbiota composition of these STI-positive patients compared with that of the non-STI group may help to identify specific bacterial taxa associated with these STIs in Taiwanese women.

We found that *L. iners* dominated the vaginal microbiome of CT-positive cases compared with the non-STI and TV-positive groups. Additionally, a higher relative abundance of *G. vaginalis* was observed in the CT-infected group than in the non-STI group. These results are consistent with previous findings, indicating that women testing positive for CT infection are more likely to have vaginal microbiomes dominated by *L. iners* than by *L. crispatus* [17] and that BV-associated microbiota increased risk of subsequent chlamydia infection [37]. It has been shown that *L. iners*-dominated vaginal microbiota is associated with increased susceptibility to CT infection due to reduced D(−) lactic acid production [38]. Characterization of vaginal bacterial communities indicated that *L. iners*-dominated CST-III were more likely to switch to CST-IV than other community states, suggesting the linking of *L. iners* to CT infections and BV [39]. Further investigations on the crosstalk between the highly enriched bacterial species and CT will be helpful to elucidate whether additional treatments such as probiotics could reduce an individual’s susceptibility to the STI.

Based on the vaginal microbiome analysis of TV-infected patients, we demonstrated that *L. iners,* the most abundant *Lactobacillus* species in the CT-positive and non-STI groups, was significantly reduced. Meanwhile, the relative abundance of GBS, *P. bivia*, *S. sanguinegens*, and *G. asaccharolytica* was enriched in the TV-positive samples. Our study parallels earlier findings, indicating that TV infection was associated with the vaginal microbiota belonging to the low-lactobacilli CST-IV group [18,40], where the presence of *Prevotella* spp. and *S. sanguinegens* was also demonstrated [19]. Additionally, it has been reported that a higher relative abundance of *G. asaccharolytica* was associated with significantly higher odds of HIV acquisition [41], suggesting a possible connection between TV-infected patients and HIV infection. It has been well known that GBS colonization in the vagina during pregnancy increases the risk of neonatal infection [42], which may result in several fetal injuries, including tissue damage, inflammation, lung and brain injury, pneumonia, meningitis, sepsis, and even fetal death [43]. However, the vaginal bacteria or other microorganisms promoting GBS colonization and adverse perinatal outcomes are far from understood. We showed for the first time that GBS is enriched in the vaginal microbiome of TV-infected women using both culture-dependent and culture-independent methods. *Lactobacillus* spp., including *L. acidophilus*, *L. paracasei* and *L. reuteri*, have been shown to prevent GBS adherence or outcompete GBS for adherence to vaginal epithelial cells [44], as well as to prevent GBS colonization in a mouse model [45]. Since *Lactobacillus* spp. are markedly reduced in the TV-positive cases compared to the CT-positive and non-STIs groups, it is likely that the inhibitory effect mediated by *Lactobacillus* spp. on GBS colonization is also decreased in this scenario. Additionally, it has been suggested that *P. bivia* may play a role in human GBS colonization [46] and is correlated with an increased risk of HIV acquisition [47]. Hence, *P. bivia* enriched in the vagina of TV-positive patients may link to enhanced GBS colonization and susceptibility to HIV infection. Together, understanding the mechanisms by which TV promotes *P. bivia* and *G. asaccharolytica* growth may provide novel strategies to reduce women’s risk of GBS and HIV infections.

Using an in vitro co-culture assay, we further proved the reciprocal interactions between TV and the associated bacterial species enriched from the microbiome analysis, including *L. iners* and GBS. We found that lactobacilli are significantly reduced after an initial interaction with TV. It has been shown that lactobacilli could be ingested by TV after 2 h of co-incubation [48]. Hence, it is likely that other specific vaginal bacteria may proliferate under this low-lactobacilli condition. On the other hand, TV that engulfs and digests bacteria may help to fulfill nutritional requirements during the establishment of an infection [49], as also supported by the enhanced growth of the parasite co-cultured with *L. iners*. We observed that GBS inhibits the growth of TV via an unknown mechanism. GBS has been shown to kill endothelial cells [50], epithelial cells [51], and fibroblasts [52] by *β*-hemolysin and induces apoptosis in macrophages via several virulence factors [53,54]. Hence, it is likely that GBS suppresses TV growth by using similar mechanisms and this hypothesis needs further investigations. On the other hand, it remains to be determined how TV promotes GBS growth, which will shed new light on the role of TV in GBS colonization and possibly reduce the risk of GBS infection.

The BD Max CT/GC/TV platform used in this study provided high sensitivity and specificity for the diagnosis of the three most common STIs at the same time [21]. However, several limitations of this study need to be considered. As few identified GC-positive cases in women with vaginitis may result in failure to evaluate gonorrhea-associated bacterial species, more GC-positive cases will be needed in future studies. Additionally, we did not exclude the samples collected from women with vaginitis pretreated with any antibiotic, which may affect the microbiota composition in the vagina. A previous study has identified several differences in the composition of the vaginal microbial communities of pregnant women compared to non-pregnant women [55]. Since our study population is comprised of non-pregnant women, the vaginal microbiota of the STIs during pregnancy may not be reflected. The effects of menopausal status and different contraceptive methods on the vaginal microbiota were also not considered in this study. Menopause has been reported to significantly influence the community structure in the vagina, and *Prevotella* spp. is strongly associated with premenopause [56]. Additionally, a previous study suggests that changes in the vaginal bacterial composition are not associated with the use of a copper intrauterine device (Cu-IUD) and a levonorgestrel intrauterine system [57]. However, another study indicated that Cu-IUD may increase colonization by BV-associated microbiota, whereas most hormonal contraception does not alter vaginal microbiota [58]. The uninterrupted use of the combined oral contraceptive pill has been demonstrated to maintain the normal microbiota and reduce the frequency of BV [59]. Hence, the vaginal microbiota may be heterogeneous and dynamically affected by the physiological states of the host.

## 5. Conclusions

Collectively, to the best of our knowledge, this is the first report investigating the prevalence of CT, TV, and GC in women with vaginitis using a molecular-based approach in Taiwan. Additionally, vaginal microbiome analysis revealed that *L. iners* and *G. vaginalis* were found to be associated with CT-positive patients, whereas GBS, *P. bivia*, *S. sanguinegens* and *G. asaccharolytica* were more abundant in TV-positive patients. Co-culture assays between TV and the associated bacteria further clarified the dynamic interactions. Future research studies could develop novel vaginal health interventions targeting the specific vaginal bacterial taxa identified from our study, elucidating the potential roles for the associated microbiota in the promotion or inhibition of common STIs.

## Figures and Tables

**Figure 1 microorganisms-09-01864-f001:**
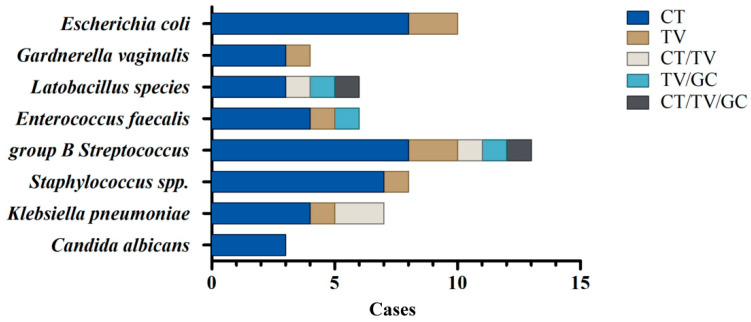
Identification of bacterial species co-infected with STIs by a culture-dependent method. Vaginal swabs for culture were plated on the suitable agar plates and medium. The co-existing microorganisms in each STI group were identified by phenotypic methods, Gram stains, and MALDI-TOF analysis.

**Figure 2 microorganisms-09-01864-f002:**
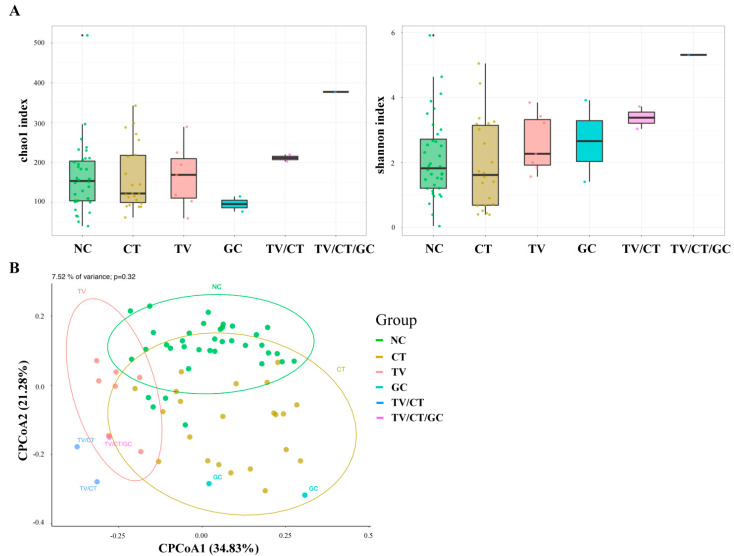
Microbial community variations between STI-positive and non-STI (NC) groups. (**A**) Bacterial α-diversity was determined by Chao and Shannon’s indices in STI-positive and NC groups. (**B**) Principal component analysis (PCoA) in STI-positive and NC groups. The individual samples were color-coded in the NC group and STI-positive patients, including CT, TV, GC, TV/CT, and TV/CT/GC.

**Figure 3 microorganisms-09-01864-f003:**
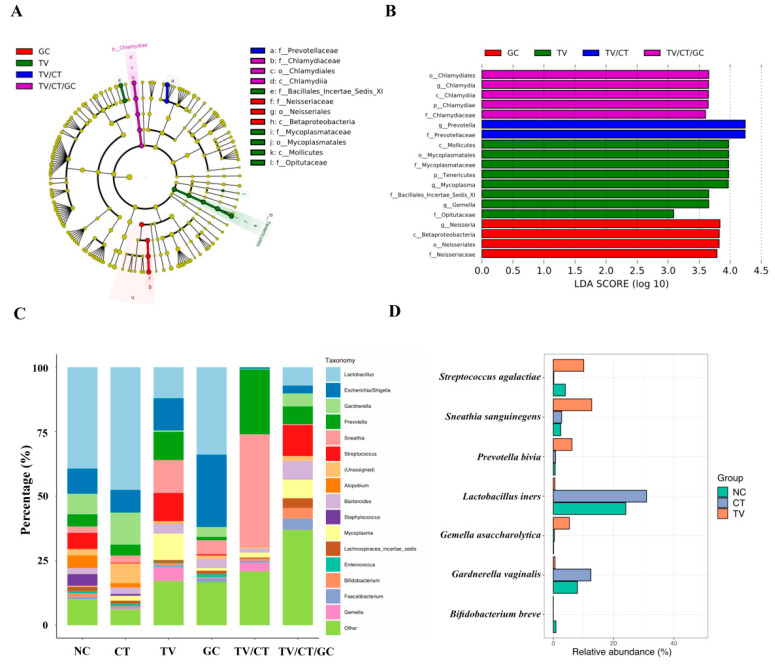
The differentially enriched bacteria in STI-positive patients compared to the non-STI (NC) group. (**A**) Cladogram generated by linear discriminant analysis effect size (LEfSe) indicated differences in taxa among STI-positive and the NC groups (**B**) Histogram of the linear discriminant analysis (LDA) scores for differentially abundant taxonomic features among the STI-positive and NC groups. (**C**) Taxonomic profiles of the most abundant bacterial genus among the STI-positive patients and NC groups. (**D**) The enriched bacterial species was highlighted in CT- and TV-positive patients compared with the NC group.

**Figure 4 microorganisms-09-01864-f004:**
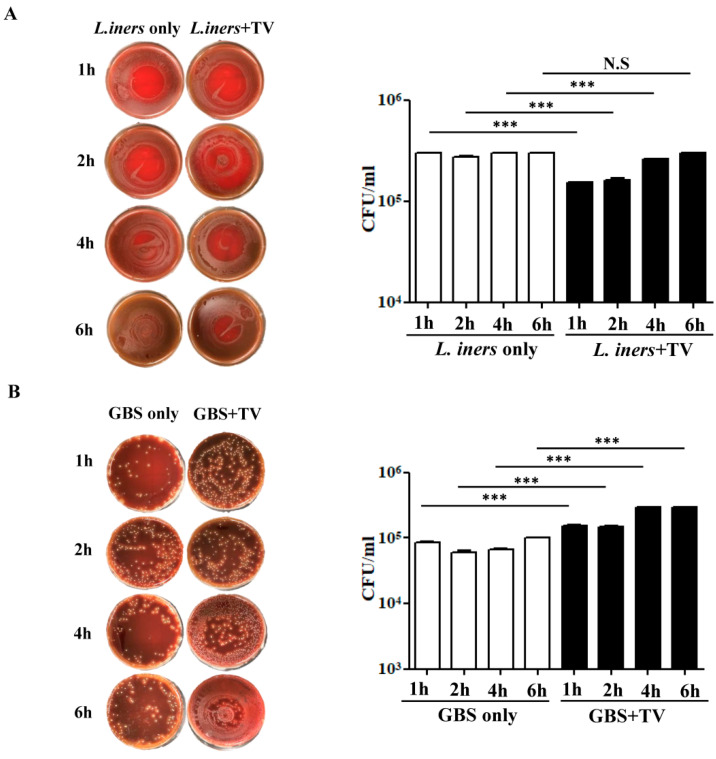
The effect of TV–bacteria interactions on the growth of specific bacteria enriched from microbiome analysis. TV (5 × 10^5^ cells/mL) was co-cultured with *L. iners* (**A**) and GBS (**B**) at a ratio of 1:10 in 1 mL of KSFM medium and the microbial culture was incubated at 37 °C for 1, 2, 4, 6 h. The growth of *L. iners* and GBS was observed after plating out the culture on blood agar plates for 24 h at 37 °C, followed by CFU counting. Right panels represent the quantitative data of CFUs from three independent experiments. *** *p* < 0.001, N.S—not significant.

**Figure 5 microorganisms-09-01864-f005:**
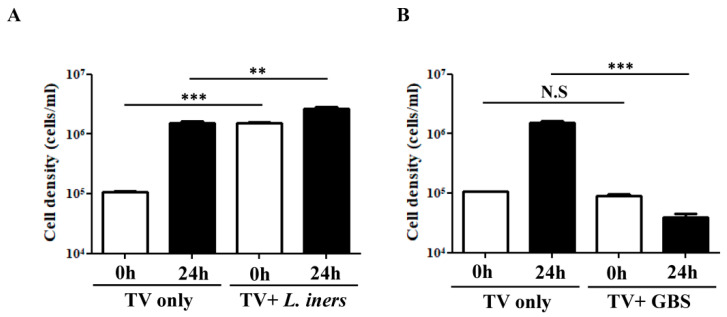
The impact of crosstalk between TV and the associated bacteria on the survival of parasites. TV survival was monitored after co-incubations of TV with *L. iners* (**A**) and GBS (**B**) at a ratio of 1:10 for 6 h in 1 mL of KSFM medium, followed by inoculation of the microbial cultures into YIS medium. The cell density of TV was monitored by using trypan blue exclusion hemocytometer counts. Quantitative data were presented as mean ± SD of three independent experiments. ** *p* < 0.01, *** *p* < 0.001, N.S—not significant.

**Table 1 microorganisms-09-01864-t001:** Demographic information, clinic type, differential diagnosis and prevalence of STIs in this study.

	Total Sample		Total STIs		CT		TV		GC	
STI Results	*n*	%	*n*	%	*n*	%	*n*	%	*n*	%
	327		48	14.6	35	10.8	7	2.2	2	0.6
Demographic characteristics										
Age, Median	39		29.1		31.6		37.1		25.2	
18 yrs and below	20	6.1	6	12.5	5	14.2	0	0	1	50
19 to 29 yrs	73	22.3	20	41.6	14	40	2	28.5	0	0
30 to 39	85	25.9	11	22.9	9	25.7	1	14.2	1	50
40 yrs. and above	149	45.7	11	22.9	7	20	4	57.1	0	0
Clinic type										
ER	62	18.7	8	16.6	7	20	1	14.2	0	0
STD clinic	98	29.8	15	31.2	11	31.4	1	14.2	1	50
OB/GYN	167	51.5	25	52	17	48.5	5	71.4	1	50
Differential diagnosis										
Vulval pruritus	18	5.5	4	8.3	4	11.4	0	0	0	0
Acute vaginitis	9	2.8	2	4.1	2	5.7	0	0	0	0
Subacute vaginitis	65	19.9	8	16.6	5	14.2	2	28.5	1	50
Cervicitis	53	16.2	8	16.6	5	14.2	2	28.5	0	0
PID	40	12.2	8	16.6	7	20	0	9	0	0
Vulvar vestibulitis	32	9.8	4	8.3	2	5.7	1	14.2	0	0
Gonorrhea	2	0.6	1	2	1	2.8	0	0	0	0
VVC	12	3.7	2	4.1	1	2.8	1	14.2	0	0
Leiomyoma	6	1.8	2	4.1	2	5.7	0	0	0	0
Trichomoniasis	1	0.3	1	2	0	0	1	14.2	0	0
Peritonitis	5	1.5	2	4.1	0	0	0	0	1	50
Atrophic vaginitis	17	5.2	0	0	0	0	0	0	0	0
Battered rape	20	6.1	6	12.5	6	17.1	0	0	0	0

STIs = Sextually transmitted infections; STD = Sextually transmitted disease; CT = *Chlamydia trachomatis*; TV = *Trichomonas vaginalis*; GC = *Neisseria gonorrhoeae*; ER = Emergency room; OB/GYN = Obstetrics and gynecology; PID = pelvic inflammatory disease; VVC = vulvovaginal candidiasis.

## Data Availability

Not applicable.

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
