# Peer review of "Vaginal Microbiota of the Sexually Transmitted Infections Caused by Chlamydia trachomatis and Trichomonas vaginalis in Women with Vaginitis in Taiwan"

_microorganisms, 2021, doi:10.3390/microorganisms9091864_

Round 1
Reviewer 1 Report
It is a very interesting and well done study. However, I recommend stratifying the data in relation to:
- pre- and post-menopause
- oral contraceptive pill
- IUD carriers
Reviewer 2 Report
In this study a total of 327 woman from Taiwan was screened for STIs using the BD Max CT/GC/TV assay. The group further characterized the vaginal ecosystem using culture dependent and culture independent methods. Using a combination of these methods such as MALDI-TOF Analysis and 16S rRNA sequencing methods they identified the different profiles of bacteria from the different groups. The researchers further performed some in vitro experiments on TV and different bacteria such as lacto bacillus iners or GBS. The authors report a prevalence for CT, TV, and GC were 10.8 %, 2.2 % and 0.6 %. They show in the in the CT-positive patients, the vaginal microbiota was dominated by L. iners, and the relative abundance of Gardnerella vaginalis (12.46 %) was also higher than that in TV-positive patients and the non-STIs group. They further show Lactobacillus spp. was significantly lower in TV-positive patients, while GBS (10.11 %), Prevotella bivia (6.19 %), Sneathia sanguinegens (12.75 %), and Gemella asaccharolytica (5.31 %) were significantly enriched. Using an in vitro co-culture assay, they demonstrated that the growth of L. iners was suppressed in the initial interaction with TV, but it may adapt and survive after longer exposure to TV.
Minor comments
line 42 change swaps to swabs
Line 124 change moist to moisture
Line 202 three hundred and twenty-seven
line 204 BDMAX assay
Line 293 Not sure where probiotic B comes from
Throughout text the bacterial species needs to be in italics
Also please get grammar looked at in the paper as at times the English is very good but sometimes the sentences needs to be rewritten.
